# The Variability of Lake Water Chemistry in the Bory Tucholskie National Park (Northern Poland)

**Mariusz Sojka [1],\*, Adam Choiński [2], Mariusz Ptak [2] and Marcin Siepak [3]**

[1]  Institute of Land Improvement, Environmental Development and Geodesy, Faculty of Environmental Engineering and Spatial Management, Poznań University of Life Sciences, Piątkowska 94, 60-649 Poznań, Poland

[2]  Institute of Physical Geography and Environmental Planning, Adam Mickiewicz University, Bogumiła Krygowskiego 10, 61-680 Poznań, Poland; choinski@amu.edu.pl (A.C.); ptakm@amu.edu.pl (M.P.)

[3]  Institute of Geology, Faculty of Geographical and Geological Sciences, Adam Mickiewicz University, Bogumiła Krygowskiego 12, 61-680 Poznań, Poland; Marcin.Siepak@amu.edu.pl

\*  Correspondence: mariusz.sojka@up.poznan.pl

**Abstract:** The paper presents the results of chemical analysis of lake waters in Bory Tucholskie National Park (BTNP). The BTNP area is unique due to its location within a single catchment and high variability in geological structure. Moreover, the lakes have different morphometric parameters, represent different hydrological types, trophic types and thermal regimes. Another unique feature is the existence of five lobelia lakes. This name comes from the Latin name of the taxon – *Lobelia dortmanna* L. which has been included in the Polish Red Data Book of Plants. The chemical analysis included 55 parameters, within macro elements (MEs), trace elements (TEs) and rare earth elements (REEs). Low concentrations of MEs, TEs and REEs confirm the absence of anthropogenic pressure. High variation of ME, TE and REE contents between individual lakes is due to different geological structure. The cluster analysis enabled lakes to be divided into six groups taking into account all analyzed water quality parameters. The lobelia lakes were characterized by the lowest concentrations of MEs and REEs, which mainly result from the small catchment area and their mainly endorheic character. The highest variability of MEs, TEs and REEs occurred in endorheic lakes, where the geological structure was dominant. The lowest variability of MEs, TEs and REEs occurred in the lakes connected by the Struga Siedmiu Jezior stream. The analysis of MEs, TEs and REEs in relation to the environmental factors and trophic, hydrologic and thermal typology allowed a better understanding of their spatial distribution in the BTNP lakes. The obtained results indicate that the values of the studied elements were generally close to the average values noted in surface waters according to the Geochemical Atlas of Europe.

**Keywords:** National Park; lake; water chemistry; macro elements (MEs); trace elements (TEs); rare earth elements (REEs)

## 1. Introduction

Lakes show great variability in their characteristics, which is a result of geological history and gradient in climate conditions [1]. The chemical composition of lake waters in natural ecosystems is related to geological and lithological structure, topography, climatic conditions, vegetation and soil properties [2–7]. Usually, chemical components are present in the aquatic ecosystems in traces and their natural concentration. It is mostly a result of a combination of bedrock and soil weathering processes [8,9]. In a natural environment lakes act as geochemical traps for wet and dry deposition compounds, as well as for those of geogenic and biogenic origin [10].

Lakes are polluted by many kinds of pollutants [11]. Markowski and Kwidzyńska [12] described the strong impacts of human activities on water chemistry. Pollution associated with urban, industrial and agricultural activities is of particular importance [6]. Monroy et al. [13] showed that intensive mining activity and urban sewage discharge have the greatest impact on trace element supply. The scale of metal contamination is primarily dependent on the distance of a lake from the source of pollution, but it is largely modified by hydrological factors [14]. Many studies on the content of macro elements (MEs) and trace elements (TEs) in lake water have been carried out around the world. However, the interest in REEs has increased significantly in recent years. Emission of rare earth elements (REEs) into natural waters is increasing, which may result in a potential risk to the environment. The REEs are accumulating in aquatic organisms, and therefore, have the potential to enter the food chain [15]. Since REEs are present in very low concentrations in natural waters, methods have been proposed to enable their measurement [16].

In Poland, there are about 7000 lakes with an area of greater or equal to 1 ha [17]. Depending on their location, lakes undergo the influence of various anthropogenic pressures [18]. Lakes located in the south of Poland in the mountain region in Tatra National Park differ in the chemical composition of waters, resulting from the geological structure and the lithology [4]. Tatra National Park lakes are also subject to various forms of anthropogenic pressure especially precipitation-related pollution (i.e., acid rain), intensive mountaineering and the effect of mountain hostel infrastructure [19,20]. The northern part of Poland is rich in coastal lakes of various connectivity with the sea. The chemical composition of intermittently closed and open lakes is diverse, while the anthropogenic impact as well as sea water intrusions determine its quality on the temporal and spatial scale, and the morphological, hydrological and anthropogenic impact makes them ecologically and chemically complex [21]. Lakes located in the central part of Poland in the lowland region are under agricultural, industrial and tourist pressure [22].

Considerable transformation of the natural environment caused by human activity necessitates some protective actions. The key issue, in this regard is the establishment of national parks where the inherent values of the natural environment are to be preserved. Such projects are carried out, among others, in relation to surface waters, which may change their condition relatively quickly. This statement can be considered in many ways, i.e., in relation to water resources [23–26] as well as to physical and chemical properties of rivers or lakes [27–31]. About 146 lakes are located in Polish natural reserves in protected areas. The analysis of water chemistry of lakes located in Polish nature reserves indicates that they are polluted [32]. In the Polish national parks, there are 95 lakes [33]. The poor condition of the lakes is related to their location within the influence of intensified human pressure. The functioning of lakes in the areas of national parks and landscape parks does not fully protect them, as pollution flows mainly together with river waters. The Bory Tucholskie National Park (BTNP) is different from the others. In the context of water circulation, most of the park area (85.5%) corresponds to the catchment area. The catchment itself has a forest character and is free of anthropogenic pressure. Thus, it can be treated as a natural catchment (quasi-natural), which from the perspective of the water cycle gives an overview of undisturbed processes that determine the functioning of lake ecosystems.

The monitoring of lake water pollution is essential in order to provide the baseline data which is very useful for the authorities for environmental management [34]. Assessment of chemical components behavior in various drainage systems encompassing a diverse range of lithologies, climates and landscapes is helpful to improve knowledge of the processes that govern its concentration and distribution [35]. In order to protect lakes and prevent their further degradation in the years 2010–2015 in Poland 493 lakes with an area of 46.3 ha to 11,340.4 ha were surveyed as part of the national monitoring program. Within the framework of chemical state monitoring, microelements As, Ba, Cr, B, Zn, Cu, Al, Mo, Se, Ag, Tl, Ti, V, Sb, F, Be, Co and Sn, as well as oil hydrocarbons, cyanides and phenols, were determined. In addition, scientists from various universities are involved in lake water quality research. The objectives of these studies are to determine the spatial and temporal distribution of water chemistry to assess the background status and estimate natural and anthropogenic

sources [27,36,37]. The aim of the research is also to investigate the role of suspended solids and debris in the process of transporting pollutants in the river system and natural and artificial reservoirs [38,39]. Various multivariate statistical methods including cluster analysis (CA), discriminant analysis (DA), factor analysis (FA) and principal component analysis (PCA) were used to explain the spatial and temporal patterns of surface water chemistry [40]. Many reports have shown that multivariate statistical methods enable better understanding of the relation between water chemistry and environmental factors [36,41–43].

The main aims of this study were as follows: (1) analysis of physical and chemical state of waters in lakes of the BTNP, (2) analysis of physical and chemical state diversity in lake waters, (3) analysis of similarities and differences in the content of MEs, TEs and REEs in lake waters, (4) analysis of relations between morphometric, trophic, hydrological and thermal conditions in relation to ME, TE and REE concentrations in lake waters. The reason for conducting research on the water chemistry of lakes located within the BTNP is the slight impact of anthropogenic factors mainly resulting from the forest use and location in one catchment area. Additionally, in the studied area there is high diversity of geological structures and morphometric, thermal, hydrological and trophic parameters of the lakes. Moreover, in the BTNP area there are unique lobelia lakes. The results obtained may serve as a point of reference in the context of research carried out by other researchers in the field of lake water chemistry.

## 2. Materials and Methods

### 2.1. Study Area

The Bory Tucholskie National Park (BTNP) was established in 1996. The BTNP covers an area of 46.13 km$^2$. The landscape of the BTNP consist of tunnel valleys of different orientation occupied by lakes and biogenic plains and extensive outwash plains and fans in the area of the ice-sheet foreland related to the last glaciation [44]. The morphology is variable due to the development of tunnel valleys, kettle holes and aeolian dunes. The BTNP is located at an altitude of 119.46 m a.s.l. near Lake Mielnica to 148.75 m a.s.l. the area located in the southern part of the BTNP. Sandy glacial sediments are the main parent material of the soils. The largest area is occupied by sands (90.2%), followed by loamy and loamy sands (7.0%). In addition, hydrogenic soils (2.8%) represented by mudslides and peats occur within the lake valley area. In the area of the BTNP, forest areas (86.8%) and surface waters (11.2%) are the dominant form of land use. A small area is occupied by arable and built-up areas (0.1% in total) and meadows and pastures (1.9%). The total length of roads in the BTNP area is 116 km, with a density of 2.53 km·km$^{-2}$. The ratio of forests in the BTNP area has increased from 75.2% to 86.8% over the last 120 years, mainly due to a reduction in the arable area from 9.8% to less than 0.1%. The average rainfall in the BTNP area is 585 mm and the average annual temperature is 7.4 °C. The main value of the BTNP is represented by 24 lakes with a total area of 5.37 km$^2$. In relation to the origin in the BTNP area, the lakes occur within the subglacial tunnel valleys and kettle holes [45]. The lakes Ostrowite, Zielone, Jeleń, Bełczak Lakes, Główka, Płęsno, Skrzynka and Mielnica connect the Struga Siedmiu Jezior stream (Figure 1), which flows into Charzykowskie Lake, and then flows down the Brda to the Vistula river. The lakes of Krzywce Wielkie, Błotko, Głuche and Krzywce Małe are connected by an artificial channel, which flows into the lake Skrzynka.

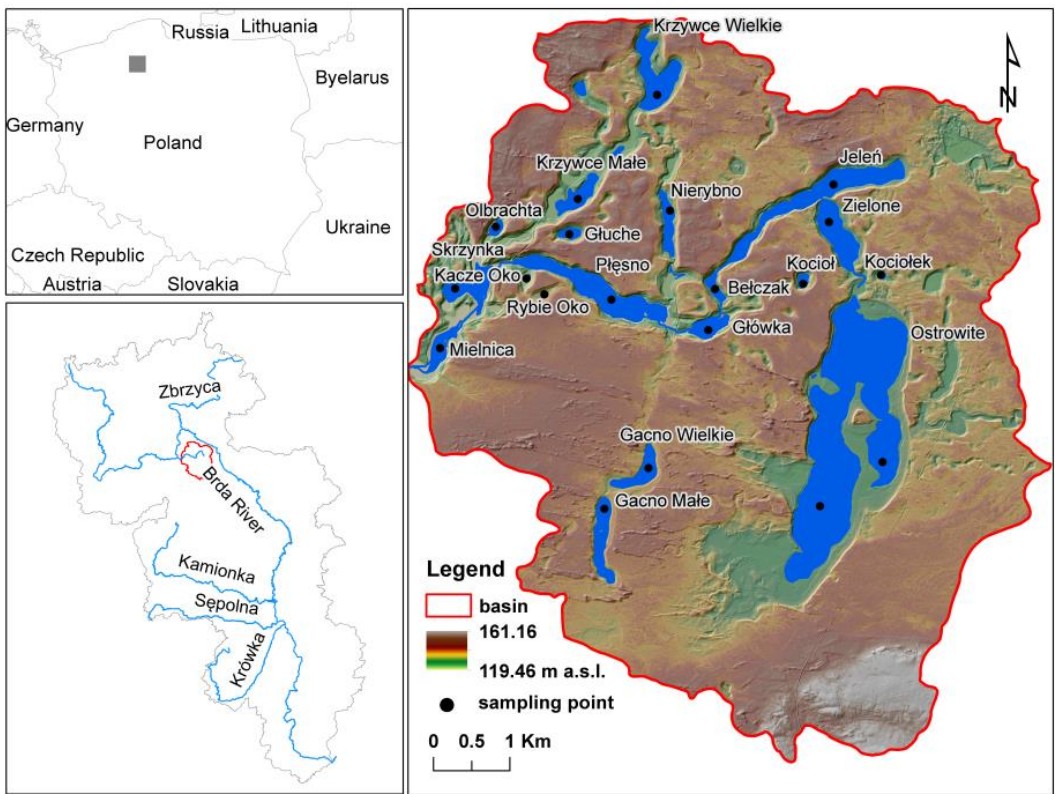

**Figure 1.** Study site location.

The hydrographic network in the area of the BTNP is poor. The total length of the rivers is 12.4 km, which gives the density of the river network of 0.27 km·km$^{-2}$. The lakes in the BTNP area can be divided into four groups: oligotrophic, mesotrophic, eutrophic and dystrophic (Table 1).

According to hydrological typology, the BTNP has flow-through, exorheic and endorheic lakes and according to thermal typology, the lakes are stratified, partially stratified and not stratified. The studied lakes are characterized by area varying from 0.3 to 280.7 ha. The volume of the lakes ranges from $5.5 \times 10^3$ m$^3$ to $29{,}989.8 \times 10^3$ m$^3$. The deepest lake in the BTNP is Ostrowite Lake with a maximum depth of 43 m, while the shallowest is Mielnica Lake with a maximum depth of 1.2 m. The lake index calculated as the ratio of the total lake catchment and lake surface area, showing the potential material supply from the catchment, varies from 6.2 to 544.2 for the Ostrowite and Bełczak lakes, respectively. One of the unique features of the BTNP, apart from its numerous lakes, is its location. The BTNP boundary in 90% coincides with the catchment area. The lack of transit rivers means that the lakes are not polluted from external sources, except for precipitation and the so-called dry and wet deposition. Another unique feature of the BTNP is the existence of five lobelia lakes. The name "lobelia lakes" comes from the Latin name of the taxon—*Lobelia dortmanna* L. This species is very rare, and therefore, it has been included in the Polish Red Data Book of Plants.

**Table 1.** Basic parameters of the lakes.

| Lake | Sample No. | Area | Volume | Mean Depth | Maximum Depth | Lake Coefficient | Trophic Type | Hydrologic Type | Stratification Type |
|---|---|---|---|---|---|---|---|---|---|
| Ostrowite | 1,2 | 280.7 | 29,989.8 | 10.7 | 43 | 6.2 | mesotrophic | Exorheic | Stratified |
| Jeleń | 3 | 48.8 | 2067.3 | 4.2 | 10.7 | 47.1 | eutrophic | Flow-through | Partly stratified |
| Płęsno | 4 | 47.8 | 2254.1 | 4.7 | 11 | 52.1 | eutrophic | Flow-through | Partly stratified |
| Krzywce Wielkie* | 5 | 26.5 | 1724.1 | 6.5 | 15 | 9.1 | mesotrophic | Exorheic | Stratified |
| Zielone | 6 | 25.5 | 2253.4 | 9 | 20.5 | 72.9 | eutrophic | Flow-through | Stratified |
| Skrzynka | 7 | 21.1 | 369.8 | 1.8 | 4.6 | 179.6 | eutrophic | Flow-through | Partly stratified |
| Gacno Małe* | 8 | 15.5 | 481.2 | 3.1 | 5.6 | 11.6 | oligotrophic | Endorheic | Non stratified |
| Gacno Wielkie* | 9 | 13.5 | 376 | 2.7 | 6.1 | 43.0 | oligotrophic | Endorheic | Non Stratified |
| Krzywce Małe | 10 | 11.8 | 621.7 | 5.3 | 10 | 51.7 | eutrophic | Flow-through | Partly stratified |
| Mielnica | 11 | 11.3 | 82.7 | 0.7 | 1.2 | 362.8 | eutrophic | Flow-through | Non stratified |
| Nierybno* | 12 | 10.1 | 260 | 2.5 | 6.3 | 13.9 | oligotrophic | Endorheic | Non stratified |
| Główka | 13 | 8 | 299.5 | 3.7 | 11 | 310.0 | eutrophic | Flow-throu13gh | Partly stratified |
| Bełczak | 14 | 4.3 | 147.2 | 3.4 | 6 | 544.2 | eutrophic | Flow-1through | Partly stratified |
| Głuche* | 15 | 3.3 | 104.5 | 3.2 | 6.5 | 15.2 | oligotrophic | Endorheic | Non stratified |
| Olbrachta | 16 | 2.6 | 67.4 | 2.6 | 3.7 | 250.0 | eutrophic | Flow-through | Non stratified |
| Kocioł | 17 | 2.5 | 170.7 | 6.8 | 11 | 12.0 | oligotrophic | Endorheic | Non stratified |
| Kociołek | 18 | 0.7 | 16 | 2.2 | 5.3 | 14.3 | dystrophic | Endorheic | Non stratified |
| Kacze Oko | 19 | 0.5 | 5.5 | 0.9 | 1.3 | 20.0 | dystrophic | Endorheic | Non stratified |
| Rybie Oko | 20 | 0.3 | 11.2 | 3.5 | 6.1 | 66.7 | dystrophic | Endorheic | Non stratified |

*2.2. Sample Collection and Preparation*

Water samples for physicochemical analysis were collected on 7–8 August 2017 from 19 lakes located in Bory Tucholskie National Park (Figure 1). Samples were taken at the deepest point of each lake, 1 m below the water surface. In the case of Lake Ostrowite, samples were taken at two points, i.e., at the deepest points of the two characteristic basins. Measurements taken directly in the field included the temperature, pH reaction and electrolytic conductivity of the water with the help of a multifunctional measuring device, Multi 350i, made by the firm WTW (Weilheim, Germany). The water samples with volume of 500 mL were collected in polyethylene bottles (HDPE) produced by Nalgene®. Samples were fixed 60% $HNO_3$ Ultrapur® (pH < 2) and $CHCl_3$ Pro Analysis® (Merck; Darmstadt, Germany). After sampling, the samples were taken to the chemical laboratory in a mobile refrigerator at a temperature of 4 ± 2.5 °C. Adequate precautions were exercised to avoid contamination of water during sampling, transport, and handling.

*2.3. Chemical Analysis*

The samples of surface water were also analyzed for their cations ($Na^+$, $NH_4^+$, $K^+$, $Ca^{2+}$, and $Mg^{2+}$) and anions ($Cl^-$, $F^-$, $NO_3^-$, $NO_2^-$, $SO_4^{2-}$, and $PO_4^{3-}$) by employing a Metrohm ion chromatograph (IC), model 881Compact IC Pro (Metrohm, Switzerland). Instrument parameters and the parameters of the analytical method have been presented separately [46,47].

Concentrations of Ag, Al, As, Ba, Be, Bi, Cd, Co, Cr, Cu, Ce, Li, Fe, Mn, Ni, Pb, Sb, Se, Sc, Sr, Sm, V, Mo, Zn, Rb, Re, La, Pr, Nd, Eu, Gd, Tb, Dy, Ho, Er, Tm, Lu, Th and U were determined by inductively coupled plasma mass spectrometry (ICP-QQQ 8800 Triple Quad, Agilent Technologies, Japan). This instrument is equipped with a MicroMist nebulizer and a Peltier-cooled (2 °C) Scott-type spray chamber for sample introduction. This instrument contains an octopole-based collision/reaction cell, located in-between two quadrupole analyzers. It was operated in a gas mode, with $O_2$ flowing at 0.3 mL min$^{-1}$ (30%) and He flowing at 5 mL min$^{-1}$. All parameters were manually optimized to achieve the best signal intensity and stability. MassHunter software for ICP-QQQ (Agilent Technologies, Japan) was used to control the instrument and to process the data [48,49]. The instrumental operating parameters are given in Table 2.

**Table 2.** ICP-QQQ operating conditions.

| Spectrometer | Agilent 8800 Triple Quad |
|---|---|
| Nebulizer | Micromist |
| Interface | Sampler and skimmer cones in Ni |
| RF power | 1550 W |
| RF matching | 1.80 V |
| Plasma flow rate (L/min) | 15 |
| Carrier gas flow (L/min) | 1.08 |
| Nebulizer pump (rps) | 0.3 |
| S/C temp (°C) | 2 |
| Sample depth (mm) | 8.0 |
| Gas flow rate | He 5.0 (mL/min) |
| | $O_2$ 0.3 (mL/min) 30 (%) |

Alkalinity was measured by in situ titration with HCl (0.1 N), using methyl orange as an indicator. Chemical oxygen demand (COD), was determined by the per manganese method. As a quality control measure, the ionic error balance was calculated. The calculated error did not exceed ±3%.

*2.4. Reagents*

For ion chromatography (IC), Merck standard solutions (Merck, Darmstadt, Germany) and CPAchem (C.P.A. Ltd. Stara Zagora, Bulgaria) were used. The mobile phase for cations and anions was made from Fluka reagents (Sigma-Aldrich, Steinheim, Switzerland). During the determination with

the technique ICP-QQQ was done using calibration curves obtained from diluted stock multi-element standard 100 µg/mL (VHG Labs, Manchester, USA). The reagents used were ultrapure, and the water was de-ionized to a resistivity of 18.2 MΩ cm in a Direct-Q$^®$ 3 Ultrapure Water System apparatus (Millipore, France). Analytical quality control was verified by the analysis of certified reference materials for lake water TMDA-51.4 (Environment Canada, Burlington, Canada) and SPS-SW2 (Spectrapure Standards as, Oslo, Norway) and high compliance with reference values was found.

## 2.5. Statistical Analysis

In the first stage of statistical analysis, lakes were divided into groups by means of cluster analysis (CA). Prior to further CA analyses, the data were transformed to logarithms log (x + 1), centred and standardized. Cluster analysis (CA) was applied to group lakes into clusters on the basis of similarities between concentration of macro elements (MEs), trace elements (TEs) and rare earth elements (REEs). Ward's method using squared Euclidean distances was used as a measure of similarity. A dendrogram was used to visualize and interpret the results of the CA. The results of clustering lakes on the basis of MEs, TEs and REEs was presented using the standardized linkage distance ($D_{link} D_{max}^{-1}$) multiplied by 100 [50]. The division into groups and subgroups was made on the basis of criteria proposed by Ptak et al. [51]. For each of the groups and subgroups characteristic concentration values were calculated: minimum value, maximum value, mean value and standard deviation. To assess the statistical differences among the water quality parameters for selected groups of lakes a non-parametric test of the variance by Kruskal–Wallis and Dunn's tests as post hoc procedures (P ≤ 0.05) was performed using the statistical software STATISTICA 13.1 PL.

In order to choose another method of statistical analysis detrended component analysis (DCA) was carried out. Detrended correspondence analysis (DCA) was done to assess the environmental factors distribution pattern. The gradient length shorter than 3.0 SD of the first axis of DCA indicates that the water quality parameters and environmental factors were in linear rather than in unimodal distribution. Ter Braak and Smilauer [52] and Glińska-Lewczuk et al. [53] suggest that in such cases the principal component analysis (PCA) is more appropriate than canonical correspondence analysis (CCA).

The relationship between environmental factors and MEs, TEs and REEs in lake water was determined by principal component analysis (PCA) with multiple scaling. PCA is an ordination method which allows the interpretation of complex correlations between analyzed water quality parameters and environmental factors and presentation of the obtained results against the background of the group selected in CA methods and lake trophic state and hydrologic and thermic conditions. The number of significant principal components was selected on the basis of the Kaiser criterion of eigenvalues higher than 1. The correlation between principal component and water quality parameters was classified according to values of >0.75, 0.75–0.50 and 0.50–0.30 proposed by [54] as strong, moderate and weak respectively. The following environmental factors were considered: lake area (Area), lake volume (Volume), lake mean and maximum depth (MeanD, MaxD) and lake coefficient (LC). Moreover, during the PCA analysis the division of lakes according to trophic typology into the following lakes was used: oligotrophic (OL), mesotrophic (MS), eutrophic (EU), and dystrophic (DT). On the basis of hydrological typology, flow-through (FT), exorheic (Ex) and endorheic lakes (En) were distinguished. Taking into account the thermal conditions, the following lakes were distinguished: stratified (S), partially stratified (PS) and not stratified (NS). Additionally, during the analysis lobelia lakes (LB) were taken into account. Moreover, during the PCA analysis the division of lakes into groups and subgroups obtained during CA was used.

## 3. Results and Discussion

### 3.1. Lakes Water Chemistry

The results revealed that in most of the sampling sites, the water was slightly alkaline (from 8.00 to 9.22). The above values usually indicate the presence of carbonates of calcium and magnesium in water [55,56]. The high pH of the water may result in the reduction of TEs and REEs [57]. The acid reaction of waters was found in eight reservoirs (from 4.33 to 6.97), with the lowest pH being found in dystrophic reservoirs. The electrical conductivity (EC) of the water samples ranged from 10.7 to 269 $\mu$S/cm. Low concentrations of basic anions and cations were found in the lake waters, and the ion abundances were typically $Ca^{2+} > Mg^{2+} > Na^+ > K^+ > NH_4^+$ (Figure 2a) and $SO_4^{2-} > Cl^- > PO_4^{3-} > NO_3^- > NO_2^-$ (Figure 2b).

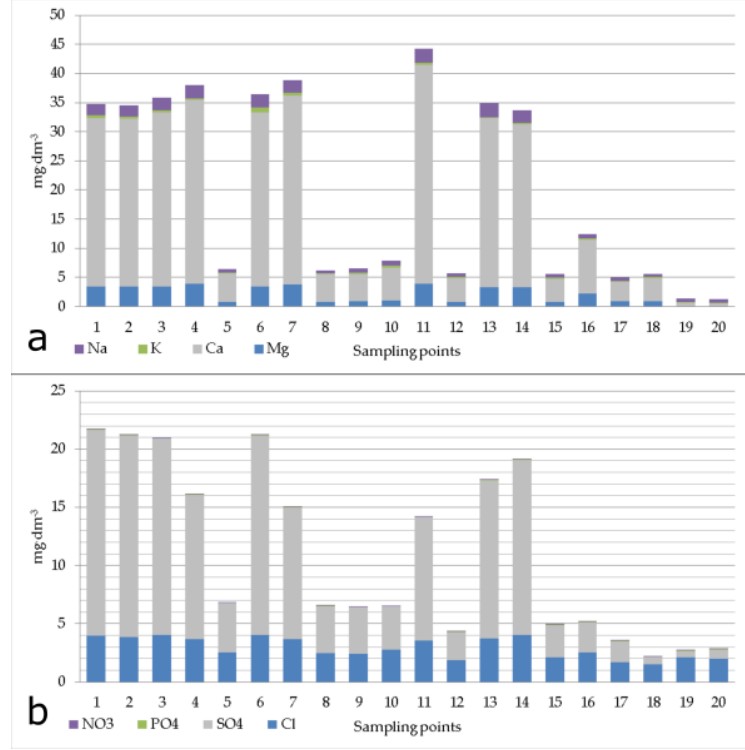

**Figure 2.** Cation (**a**) and anion (**b**) concentrations in the Bory Tucholskie National Park lakes (Sampling point number according to Table 1).

The results of the chemical analyses of water samples for macro elements (MEs), trace elements (TEs) and rare earth elements (REEs) are presented in Tables A1–A3. Maximum concentrations of MEs, TEs and REEs obtained during the study were compared with values from the European Geochemical Atlas recorded in surface waters of Europe, covering 26 countries, including Poland [58]. ME, TE and REE concentrations presented in the Geochemical Atlas of Europe have right-skewed distribution, i.e., median values are lower than the mean value. The results indicate that the maximum concentrations of $F^-$, $Cl^-$, $Ca^{2+}$, $Mg^{2+}$, $Na^+$, $K^+$, Sr, Cr, Ni, Ba, V, Co, Dy and U recorded in the BTNP lakes were higher than the minimum value and lower than the median value from the Geochemical Atlas of Europe. Fe, $HCO_3^-$, $SO_4^{2-}$, Li, Se, Rb and most of the REEs (La, Ce, Pr, Nd, Sm, Eu, Gd, Er) were higher than the median and lower than the mean value recorded in surface waters in Europe. In other cases, the maximum concentrations were higher than the mean value and only in the case of Bi exceeded the maximum value given by Salminen et al. [58]. In areas without a direct impact of anthropogenic factors, the content of MEs, TEs and REEs depends on the geological structure and the weathering process. Wang et al. [59] suggested that a large amount of TEs may originate from

atmospheric deposition and weathering of background soils. Similarly, for REEs, their surface water content may also be affected by dry and wet deposition of REEs from the atmosphere (dust, particles, rain, snow). The content of REE in the atmosphere is dependent on anthropogenic factors, especially near cities [60]. The analysis showed that the concentrations of individual elements in the lake waters were characterized by unequal variability. Average variability of parameters in ME and TE groups was at a similar level, whereas parameters in the REE group were characterized by the highest variability. Very high variability in the ME group was observed for pH and alkalinity, $HCO_3^-$, hardness, electrolytic conductivity and concentrations of $Ca^{2+}$, Fe and Mn, in the TE group Co, Mo, Sr, Al and Bi and among the REEs, La, Pr, Sm, Tb, Dy, Ho and Tm. On the other hand, the lowest variation was observed for $NO_3^-$, $Cl^-$, $F^-$, Se and Rb concentrations. The variability of other parameters was at an average or high level. Taking into account that the analyzed lakes are located within a relatively small area of about 46.13 km$^2$, none of the recorded concentrations have very low variability. The variability of MEs, TEs and REEs results from the diverse geological structure. As a result of the weathering process, REEs are easily activated and transferred from the rock environment to water. The content of REEs in surface waters (rivers and streams) varies considerably. Studies indicate that the content of REEs in river waters is mainly controlled by two factors, namely organic matter content and water pH [60].

### 3.2. Statistical Analysis

Cluster analysis was carried out in order to show similarities and differences in the chemical composition of waters in lakes. CA analysis was performed separately for MEs, TEs and REEs. On the basis of the ME analysis, the lakes were divided into two main groups, MEs1 and MEs2, within which two subgroups were separated (Figure 3).

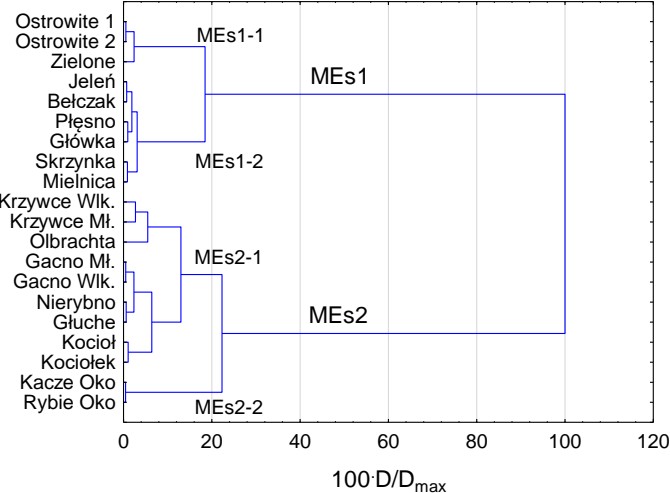

**Figure 3.** Dendrogram showing clustering of lakes on the basis of macro elements (MEs) concentration.

The MEs1 group includes the Ostrowite and Zielone lakes and Jeleń, Bełczak, Płęsno, Główka, Skrzynka and Mielnica. The MEs1-1 subgroup contains the Ostrowite and Zielone lakes, which are located in the near vicinity. According to trophic typology, these are mesotrophic and eutrophic reservoirs, respectively, where stratification occurs. Moreover, there is a hydrological connection between the lakes; the reservoirs are connected by the Struga Siedmiu Jezior stream. In the MEs1-2 subgroup, there are eutrophic and flow lakes, mainly partially stratified. These reservoirs are connected by the Struga Siedmiu Jezior stream, which continuously provides water. In the second group of MEs2 the MEs content was more varied, which results from the lack of direct hydrological connection or periodic hydrological connection (Krzywce Wielkie, Krzywce Małe, Olbrachta). The MEs2-1 subgroup includes nine lakes, mainly oligotrophic, non-stratified and endorheic reservoirs. In the MEs2-1 subgroup, high similarity of macroelement concentrations was observed in the lakes of Krzywce

Wielkie, Krzywce Małe and Olbrachta, which are located in one tunnel valley and are additionally connected by an artificial channel periodically supplying water. The two smallest lakes, Kacze Oko and Rybie Oko, are classified in the MEs2-2 subgroup; they are dystrophic, non-stratified and endorheic lakes. Analysis of concentrations of individual macroelements of separated waters showed that MEs1 had higher pH, EC, Alkalinity, Hardness, $HCO_3^-$ and mean concentrations of $Cl^-$, $SO_4^{2-}$, $K^+$, $Na^+$, $Mg^{2+}$ and $Ca^{2+}$. In the MEs2 group very high concentrations of Mn and Fe were recorded. Concentrations of $PO_4^{3-}$, $NO_3^-$, $NO_2^-$ and $NH_4^+$ were at very different levels in individual groups. Higher concentrations were generally found in MEs1-2 and MEs2-2 subgroups. Cluster analysis carried out on the basis of TEs allowed the lakes to be divided into TEs1 and TEs2 groups as well. The TEs2 group was generally characterized by higher concentrations of Al, Ni, Cu, Se, Rb, Sb and Pb than the TEs1 group. Higher concentrations of Li, Sc, V, Sr, Mo, Ba and Bi were recorded in the TEs1 group. Within the TEs1 group two subgroups were distinguished. The TEs1-1 subgroup includes sample points located in Ostrowite Lake. Seven lakes were included in the second subgroup of TEs1-2 (Figure 4).

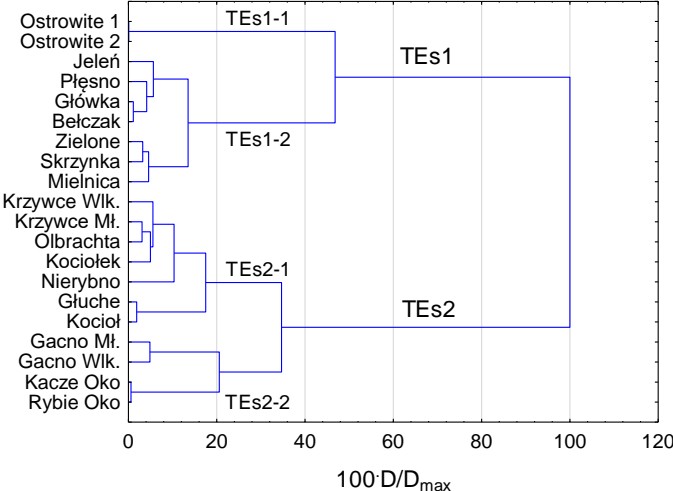

**Figure 4.** Dendrogram showing clustering of lakes on the basis of trace elements (TEs) concentration.

The Struga Siedmiu Jezior stream flows through the lakes of the TEs1-2 group. The lakes included in this group are mainly eutrophic-partially stratified. The TEs2 group includes 11 lakes, seven of which belong to the TEs2-1 subgroup and four to the TEs2-2 subgroup. Among the lakes included in the second group are the endorheic and flow-through lakes: lakes Krzywce Wielkie, Krzywce Małe and Olbrachta. The TEs2 group consisted mainly of non-stratified reservoirs. The TEs2 group was characterized by greater diversity of trace elements content than the TEs1 group. This is mainly due to the lack of hydraulic connection, and the content of trace elements in waters is determined mainly by the geological structure and supplied with groundwater. In flow-through lakes, trace elements can be transported by suspended sediments. The CA analysis carried out in relation to REEs allowed the lakes to be divided into three groups: REEs1, REEs2 and REEs3 (Figure 5).

The REEs1 group included Ostrowite Lake and the REEs2 group included the three lakes Gacno Małe, Kacze Oko and Rybie Oko, which are located on the left bank of the Struga Siedmiu Jezior stream. The lakes are endorheic reservoirs in which there is no thermal stratification of waters. In terms of trophic state, Kacze Oko and Rybie Oko are dystrophic lakes and Gacno Małe is an oligotrophic lake. The third group consisted of 15 lakes divided into two subgroups: REEs3-1 and REEs3-2. Less diverse concentrations of REEs were observed for the REEs3-2 group, which included nine lakes, mainly endorheic lakes or lakes located on periodic streams. Zielone Lake, located just below Ostrowite Lake, is an exception in this group. The trophic state of lakes and thermal conditions vary significantly in this group. In the REEs3-1 group there are only flow-through lakes connected by the Struga Siedmiu Jezior

stream. According to trophic typology, these lakes are classified as eutrophic lakes, whereas in terms of thermal conditions they are mainly stratified lakes. Analysis of REE concentrations in separated groups showed that the highest concentrations were found in Ostrowite Lake. In the remaining groups the highest mean concentrations were found in lakes included in the REEs3-2 group, then REEs2, and the lowest concentrations were found in lakes included in the REEs3-1 group.

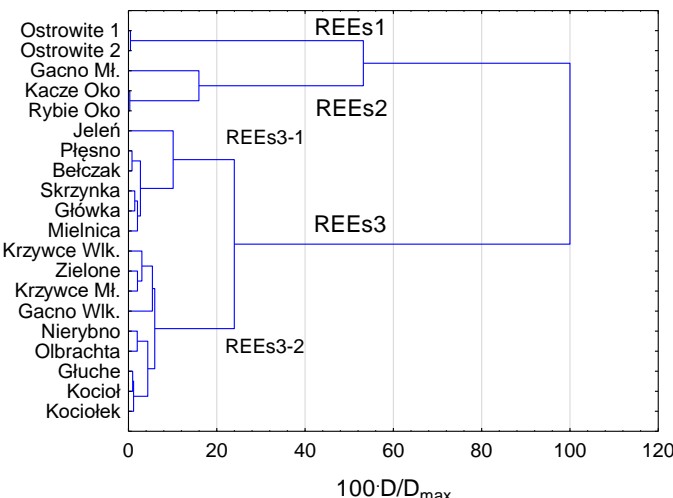

**Figure 5.** Dendrogram showing clustering of lakes on the basis of rare earth elements (REEs) concentration.

Summarizing the cluster analysis according to MEs, TEs and REEs, lakes can be divided into six groups: I Lake Ostrowite, II Lake Zielone, III lakes Jeleń, Bełczak, Główka, Płęsno, Skrzynka and Mielnica, IV lakes Krzywce Wielkie, Krzywce Małe, Nierybno, Głuche, Olbrachta, Kocioł and Kołek, V lakes Kacze Oko and Rybie Oko and VI lakes Gacno Wielkie and Gacno Małe. The first group includes Lake Ostrowite, the functioning of which is independent of other lakes. The deep lake bowl is mainly supplied with groundwater. Group II includes Lake Zielone, which is a buffer between Lake Ostrowite and other lakes that are part of the river-and-lakes system of the Struga Siedmiu Jezior. The lake has a small surface area and an average depth of approximately 9.0 m, and the waters are fully stratified. The lakes Jeleń, Bełczak, Główka, Płęsno, Skrzynka and Mielnica are lakes where there is no full stratification and their average depth ranges from 0.7 to 4.7 m (Group III). These lakes are also exposed to a high supply of matter from the catchment; the lake coefficient varies from 47.1 to 544.2. The lakes of group V, depending on the type of analyzed water quality parameters, changed their membership in particular groups.

The subject of the next analysis was to assess the variability of concentrations of analyzed water quality parameters divided into groups determined by the CA method. The analysis for macro elements was performed for the four subgroups MEs1-1 (N = 3), MEs1-2 (N = 6), MEs2-1 (N = 9), MEs2-2 (N = 2), trace elements for the four groups TEs1-1 (N = 2), TEs1-2 (N = 7), TEs2-1 (N = 7), TEs2-2 (N = 4) and for rare earth elements for the four groups REEs1 (N = 2), REEs2 (N = 3), REEs3-1 (N = 6) and REEs3-2 (N = 9). For each group of lakes, the values of characteristic concentrations—minimum, maximum, mean and standard deviation—were calculated and are presented in Tables 3–5.

**Table 3.** Characteristic concentrations of macro elements (MEs) in the lakes of the Bory Tucholskie National Park according to cluster analysis (CA) groups.

| Parameters | Unit | Group | | | | |
|---|---|---|---|---|---|---|
| | | MEs1-1 | MEs1-2 | MEs2-1 | MEs2-2 | MEs |
| pH value | pH | 8.36–8.41 | 8.20–8.69 | 5.22–9.22 | 4.33–4.46 | 4.33–9.22 |
| | | 8.38–0.03 | 8.47–0.08 | 6.79–1.57 | 4.40–0.09 | 7.29–1.64 |
| EC | μS/cm | 224.0–238.0 | 226.0–269.0 | 10.7–44.2 | 23.2–27.6 | 10.7–269.0 |
| | | 229.3–7.57 | 242.3–16.07 | 27.4–12.32 | 25.4–3.11 | 121.9–108.4 |
| Alk. | mval/L | 1.40–1.50 | 1.40–2.10 | 0.20–0.60 | 0.00–0.00 | 0.0–2.10 |
| | | 1.43–0.06 | 1.68–0.25 | 0.27–0.13 | 0.00–0.00 | 0.84–0.73 |
| Hard. | mg CaCO$_3$/L | 85.7–88.7 | 83.4–109.7 | 12.2–32.2 | 1.60–1.84 | 1.60–109.7 |
| | | 86.9–1.59 | 93.1–9.54 | 16.5–6.13 | 1.72–0.17 | 48.6–40.2 |
| HCO$_3^-$ | mg/L | 85.4–91.5 | 85.4–128.1 | 12.2–36.6 | 0.00–0.00 | 0.0–128.1 |
| | | 87.4–3.52 | 102.7–15.2 | 16.3–8.07 | 0.00–0.00 | 51.2–44.6 |
| F$^-$ | mg/L | 0.04–0.05 | 0.05–0.07 | 0.04–0.07 | 0.03–0.04 | 0.03–0.07 |
| | | 0.05–0.01 | 0.06–0.01 | 0.05–0.01 | 0.04–0.01 | 0.06–0.01 |
| Cl$^-$ | mg/L | 3.84–40.5 | 3.57–4.06 | 1.53–2.80 | 2.02–2.11 | 1.53–4.06 |
| | | 3.95–0.11 | 3.79–0.20 | 2.21–0.44 | 2.07–0.06 | 2.93–0.90 |
| SO$_4^{2-}$ | mg/L | 17.1–17.7 | 10.6–16.9 | 0.63–4.29 | 0.052–0.78 | 0.52–17.7 |
| | | 17.4–0.34 | 13.30–2.35 | 2.92–1.21 | 0.65–0.18 | 7.97–6.53 |
| PO$_4^{3-}$ | mg/L | 0.05–0.07 | 0.03–0.07 | 0.01–0.06 | 0.06–0.07 | 0.01–0.07 |
| | | 0.06–0.01 | 0.05–0.01 | 0.03–0.02 | 0.07–0.01 | 0.04–0.02 |
| NH$_4^+$ | mg/L | 0.11–0.16 | 0.32–0.44 | 0.07–0.38 | 0.20–0.23 | 0.07–0.44 |
| | | 0.13–0.03 | 0.39–0.05 | 0.15–0.10 | 0.21–0.02 | 0.23–0.13 |
| NO$_2^-$ | mg/L | 0.001–0.003 | 0.004–0.006 | 0.002–0.006 | 0.005–0.006 | 0.001–0.006 |
| | | 0.002–0.001 | 0.005–0.001 | 0.004–0.001 | 0.006–0.001 | 0.004–0.001 |
| NO$_3^-$ | mg/L | 0.005–0.007 | 0.008–0.011 | 0.004–0.010 | 0.010–0.011 | 0.004–0.011 |
| | | 0.006–0.001 | 0.010–0.001 | 0.006–0.002 | 0.011–0.001 | 0.008–0.002 |
| K$^+$ | mg/L | 0.39–0.80 | 0.23–0.47 | 0.07–0.36 | 0.11–0.14 | 0.07–0.80 |
| | | 0.53–0.23 | 0.33–0.08 | 0.21–0.10 | 0.13–0.02 | 0.29–0.16 |
| Na$^+$ | mg/L | 1.96–2.26 | 2.12–2.37 | 0.44–0.86 | 0.57–0.57 | 0.44–2.37 |
| | | 2.07–0.16 | 2.25–0.10 | 0.63–0.13 | 0.57–0.01 | 1.33–0.81 |
| Mg$^{2+}$ | mg/L | 0.39–3.43 | 3.30–3.87 | 0.80–2.27 | 0.09–0.12 | 0.09–3.87 |
| | | 3.40–0.02 | 3.59–0.27 | 1.03–0.47 | 0.11–0.03 | 2.06–1.43 |
| Ca$^{2+}$ | mg/L | 28.8–29.9 | 27.9–37.6 | 3.32–9.16 | 0.50–0.53 | 0.50–37.6 |
| | | 29.2–0.60 | 31.4–3.45 | 4.93–1.73 | 0.52–0.02 | 16.1–13.8 |
| Fe | mg/L | 3.16–3.37 | 5.59–39.6 | 7.63–93.7 | 241.6–252.9 | 3.16–252.9 |
| | | 3.27–0.11 | 18.7–13.2 | 37.6–29.4 | 247.3–8.01 | 47.8–72.3 |
| Mn | mg/L | 1.64–1.87 | 7.45–50.6 | 4.93–78.6 | 58.9–68.3 | 1.64–78.6 |
| | | 1.75–0.12 | 25.6–17.9 | 42.7–29.3 | 63.6–6.61 | 33.5–27.6 |

**Table 4.** Characteristic concentrations of trace elements (TEs) in the lakes of the Bory Tucholskie National Park according to CA groups.

| Parameters | Unit | Group | | | | |
|---|---|---|---|---|---|---|
| | | TEs1-1 | TEs1-2 | TEs2-1 | TEs2-2 | TEs |
| Li | | 2.70–2.78 | 1.18–1.90 | 0.36–0.75 | 0.41–0.84 | 0.3–2.78 |
| | | 2.74–0.05 | 1.62–0.29 | 0.52–0.13 | 0.62–0.21 | 1.15–0.77 |
| Be | | 0.10–0.10 | 0.01–0.09 | 0.02–0.03 | 0.03–0.05 | 0.01–0.10 |
| | | 0.10–0.01 | 0.04–0.03 | 0.02–0.01 | 0.04–0.01 | 0.04–0.03 |
| Al | | 2.25–2.31 | 2.58–28.2 | 7.65–66.9 | 72.8–84.8 | 2.25–84.8 |
| | | 2.28–0.05 | 9.90–8.64 | 23.0–20.1 | 79.0–5.19 | 27.6–30.1 |
| Sc | | 0.20–0.21 | 0.22–0.30 | 0.04–0.11 | 0.03–0.09 | 0.03–0.30 |
| | | 0.21–0.01 | 0.25–0.02 | 0.06–0.02 | 0.06–0.03 | 0.14–0.09 |
| V | | 0.23–0.25 | 0.21–0.36 | 0.04–0.30 | 0.10–0.25 | 0.04–0.36 |
| | | 0.24–0.02 | 0.28–0.06 | 0.17–0.09 | 0.18–0.06 | 0.22–0.09 |

| Parameters | Unit | Group | | | | |
|---|---|---|---|---|---|---|
| | | **TEs1-1** | **TEs1-2** | **TEs2-1** | **TEs2-2** | **TEs** |
| Cr | | 0.15–0.16 | 0.04–0.06 | 0.03–0.11 | 0.05–0.22 | 0.03–0.22 |
| | | 0.15–0.01 | 0.05–0.01 | 0.07–0.03 | 0.14–0.09 | 0.09–0.06 |
| Co | | 0.04–0.04 | 0.02–0.05 | 0.01–0.05 | 0.06–0.13 | 0.01–0.13 |
| | | 0.04–0.00 | 0.03–0.01 | 0.03–0.01 | 0.10–0.03 | 0.04–0.03 |
| Ni | | 0.11–0.11 | 0.06–0.44 | 0.13–0.42 | 0.10–0.35 | 0.06–0.44 |
| | | 0.11–0.00 | 0.22–0.15 | 0.27–0.12 | 0.23–0.10 | 0.23–0.12 |
| Cu | | 0.47–0.49 | 0.43–2.51 | 0.90–2.52 | 1.01–2.45 | 0.43–2.52 |
| | | 0.48–0.01 | 1.20–0.73 | 1.50–0.67 | 1.74–0.80 | 1.34–0.73 |
| Zn | | 15.0–15.6 | 1.37–10.5 | 3.39–11.3 | 10.4–14.4 | 1.37–15.6 |
| | | 15.3–0.44 | 4.59–2.90 | 6.39–2.73 | 12.3–1.76 | 7.84–4.50 |
| As | | 0.80–0.82 | 0.67–1.38 | 0.46–2.04 | 0.65–1.27 | 0.46–2.04 |
| | | 0.81–0.01 | 1.01–0.27 | 1.08–0.56 | 0.90–0.29 | 0.99–0.38 |
| Se | | 0.26–0.27 | 0.09–0.24 | 0.28–0.38 | 0.18–0.41 | 0.09–0.41 |
| | | 0.27–0.00 | 0.18–0.06 | 0.33–0.04 | 0.29–0.12 | 0.27–0.09 |
| Rb | | 0.64–0.65 | 0.42–0.60 | 0.77–1.17 | 0.47–0.83 | 0.42–1.17 |
| | | 0.64–0.01 | 0.51–0.07 | 0.99–0.14 | 0.65–0.19 | 0.72–0.24 |
| Sr | | 78.4–78.9 | 68.1–91.7 | 1.73–30.5 | 3.35–7.06 | 1.73–91.7 |
| | | 78.6–0.37 | 78.2–9.28 | 9.17–9.91 | 5.16–1.92 | 39.5–36.9 |
| Mo | | 0.61–0.63 | 0.21–0.36 | 0.06–0.14 | 0.05–0.12 | 0.05–0.63 |
| | | 0.62–0.01 | 0.28–0.06 | 0.09–0.03 | 0.08–0.03 | 0.21–0.17 |
| Ag | | 0.09–0.10 | 0.01–0.03 | 0.01–0.04 | 0.02–0.03 | 0.01–0.10 |
| | | 0.09–0.00 | 0.02–0.01 | 0.03–0.01 | 0.02–0.01 | 0.03–0.02 |
| Cd | | 0.03–0.03 | 0.01–0.02 | 0.01–0.05 | 0.05–0.08 | 0.01–0.08 |
| | | 0.03–0.00 | 0.02–0.01 | 0.03–0.02 | 0.06–0.01 | 0.03–0.02 |
| Sb | | 0.17–0.18 | 0.07–0.16 | 0.11–0.44 | 0.12–0.29 | 0.07–0.44 |
| | | 0.18–0.00 | 0.10–0.04 | 0.23–0.11 | 0.20–0.08 | 0.17–0.09 |
| Ba | | 5.36–5.42 | 5.23–8.73 | 0.93–4.15 | 0.97–4.62 | 0.93–8.73 |
| | | 5.39–0.05 | 6.10–1.21 | 2.28–1.05 | 2.79–1.87 | 4.03–2.14 |
| Pb | | 0.49–0.53 | 0.39–0.87 | 0.43–0.79 | 0.85–1.64 | 0.39–1.64 |
| | | 0.51–0.02 | 0.52–0.17 | 0.57–0.13 | 1.12–0.35 | 0.66–0.30 |
| Bi | | 0.32–0.33 | 0.04–0.11 | 0.02–0.06 | 0.03–0.04 | 0.02–0.33 |
| | | 0.32–0.01 | 0.06–0.03 | 0.04–0.02 | 0.04–0.01 | 0.07–0.09 |

**Table 5.** Characteristic concentrations of rare earth elements (REEs) in the lakes of the Bory Tucholskie National Park according to CA groups.

| Parameters | Unit | Group | | | | |
|---|---|---|---|---|---|---|
| | | **REEs1** | **REEs2** | **REEs3-1** | **REEs3-2** | **REEs** |
| La | | 0.028–0.030 | 0.006–0.012 | 0.002–0.022 | 0.027–0.056 | 0.002–0.056 |
| | | 0.029–0.001 | 0.009–0.002 | 0.010–0.006 | 0.038–0.015 | 0.016–0.013 |
| Ce | | 0.030–0.031 | 0.008–0.017 | 0.007–0.040 | 0.071–0.082 | 0.007–0.82 |
| | | 0.031–0.001 | 0.012–0.003 | 0.023–0.011 | 0.075–0.006 | 0.028–0.022 |
| Pr | | 0.015–0.016 | 0.002–0.008 | 0.001–0.006 | 0.009–0.009 | 0.001–0.016 |
| | µg/L | 0.016–0.001 | 0.004–0.002 | 0.002–0.002 | 0.009–0.000 | 0.005–0.005 |
| Nd | | 0.019–0.019 | 0.002–0.012 | 0.003–0.019 | 0.029–0.046 | 0.002–0.046 |
| | | 0.019–0.000 | 0.007–0.005 | 0.011–0.006 | 0.036–0.009 | 0.014–0.011 |
| Sm | | 0.009–0.010 | 0.002–0.008 | 0.001–0.008 | 0.006–0.018 | 0.001–0.018 |
| | | 0.009–0.000 | 0.004–0.002 | 0.003–0.003 | 0.014–0.007 | 0.006–0.005 |
| Eu | | 0.018–0.019 | 0.003–0.018 | 0.001–0.011 | 0.006–0.012 | 0.001–0.019 |
| | | 0.019–0.001 | 0.010–0.005 | 0.004–0.003 | 0.008–0.003 | 0.008–0.006 |
| Gd | | 0.010–0.011 | 0.001–0.007 | 0.001–0.009 | 0.009–0.019 | 0.001–0.019 |
| | | 0.011–0.001 | 0.004–0.002 | 0.005–0.003 | 0.012–0.006 | 0.006–0.004 |

**Table 5.** *Cont.*

| Parameters | Unit | Group | | | | |
|---|---|---|---|---|---|---|
| | | REEs1 | REEs2 | REEs3-1 | REEs3-2 | REEs |
| Tb | | 0.012–0.013 | 0.001–0.008 | 0.001–0.003 | 0.002–0.005 | 0.001–0.013 |
| | | 0.012–0.000 | 0.003–0.003 | 0.001–0.001 | 0.003–0.001 | 0.003–0.004 |
| Dy | | 0.007–0.008 | 0.001–0.003 | 0.001–0.003 | 0.001–0.006 | 0.001–0.008 |
| | | 0.008–0.000 | 0.002–0.001 | 0.001–0.001 | 0.004–0.003 | 0.002–0.002 |
| Ho | | 0.013–0.014 | 0.001–0.007 | 0.001–0.004 | 0.002–0.006 | 0.001–0.014 |
| | | 0.013–0.001 | 0.003–0.002 | 0.002–0.001 | 0.004–0.002 | 0.004–0.004 |
| Er | | 0.011–0.013 | 0.002–0.007 | 0.001–0.008 | 0.004–0.006 | 0.001–0.013 |
| | | 0.012–0.001 | 0.004–0.002 | 0.004–0.002 | 0.005–0.001 | 0.005–0.003 |
| Tm | | 0.017–0.018 | 0.001–0.006 | 0.001–0.004 | 0.004–0.005 | 0.001–0.018 |
| | | 0.017–0.000 | 0.003–0.002 | 0.002–0.001 | 0.004–0.001 | 0.004–0.005 |
| Lu | | 0.011–0.012 | 0.002–0.006 | 0.002–0.006 | 0.002–0.003 | 0.002–0.012 |
| | | 0.011–0.001 | 0.003–0.002 | 0.003–0.001 | 0.003–0.001 | 0.004–0.003 |
| Th | | 0.035–0.036 | 0.011–0.026 | 0.003–0.021 | 0.007–0.021 | 0.003–0.036 |
| | | 0.036–0.001 | 0.015–0.006 | 0.010–0.006 | 0.016–0.008 | 0.015–0.009 |
| U | | 0.139–0.142 | 0.081–0.159 | 0.003–0.077 | 0.005–0.006 | 0.003–0.159 |
| | | 0.140–0.002 | 0.127–0.031 | 0.014–0.024 | 0.006–0.001 | 0.059–0.064 |
| Re | | 0.214–0.278 | 0.249–0.392 | 0.036–0.249 | 0.142–0.357 | 0.036–0.392 |
| | | 0.246–0.046 | 0.315–0.061 | 0.126–0.072 | 0.285–0.124 | 0.219–0.112 |

Then, in order to analyze the differences in the concentrations of parameters determined in the groups of lakes, the non-parametric tests of variance Kruskal–Wallis and Dunn's tests were conducted as post hoc procedures. The results of the analysis of differences are presented in Table 6.

The results show that in the groups determined with the CA method there were statistically significant differences for all MEs. Among TEs no significant differences were observed for Ni, Cu, V, As, Cd and Ag, and among REEs for Lu and Th.

In the next stage of statistical analysis, the DCA analysis was carried out separately for MEs, TEs and REEs. This analysis showed that gradient length of the first axis for MEs, TEs and REEs was 0.85, 0.65 and 0.63, respectively. These values were lower than 3.0, which indicates that the parameters were in linear distribution. For this type of data, further analysis was carried out using the PCA method.

**Table 6.** Statistical significance of differences ($p \leq 0.05$) in the concentrations of MEs, TEs and REEs in the groups of lakes identified according to the CA method.

| Class | MEs | TEs | REEs |
|---|---|---|---|
| CA Group | pH, EC**, Alkalinity***, Hardness**, HCO$_3$***, F, Cl**, SO$_4$***, PO$_4$, NH$_4$*, NO$_2$, NO$_3$, K, Na**, Mg**, Ca**, Fe*, Mn | Li**, Be, Al**, Sc**, Cr, Se, Co, Zn*, Se*, Rb**, Sr**, Mo**, Sb, Ba**, Pb, Bi | La, Ce*, Pr*, Nd*, Sm, Eu, Gd, Tb, Dy, Ho, Er, Tm, U**, Re* |

*—$p \leq 0.01$ **—$p \leq 0.005$, ***—$p \leq 0.001$.

The results of the linear ordination method PCA for lakes (represented by circle), concentrations of MEs, TEs and REEs (represented by thin dark blue arrows), quantitative environmental variables such as area, volume, mean and maximum depth (represented by thick red arrows) and nominal variables representing lake nutrient status, thermic and hydrological condition and belonging to a group designated by CA methods (represented by red triangles) are presented in Figure 6. The projection of the sample points (lakes) perpendicular to arrows enables the lake to be ordered according to concentration of selected parameters in the water. The lakes projecting further from zero in the direction of the arrow are predicted to have above-average concentration, while the lakes projecting in the opposite direction are predicted to have below-average values. Analogously, an analysis of the content of individual elements in the lake waters characterized by a specific trophic, thermal and hydrological

state and in the groups separated by the CA method was carried out. The presented graphs also allow for evaluation of the correlation between individual elements; when the arrows point in the same direction the parameters are positively correlated with each other, when they point in the opposite direction the correlation is negative and when they intersect at a right angle there is no correlation.

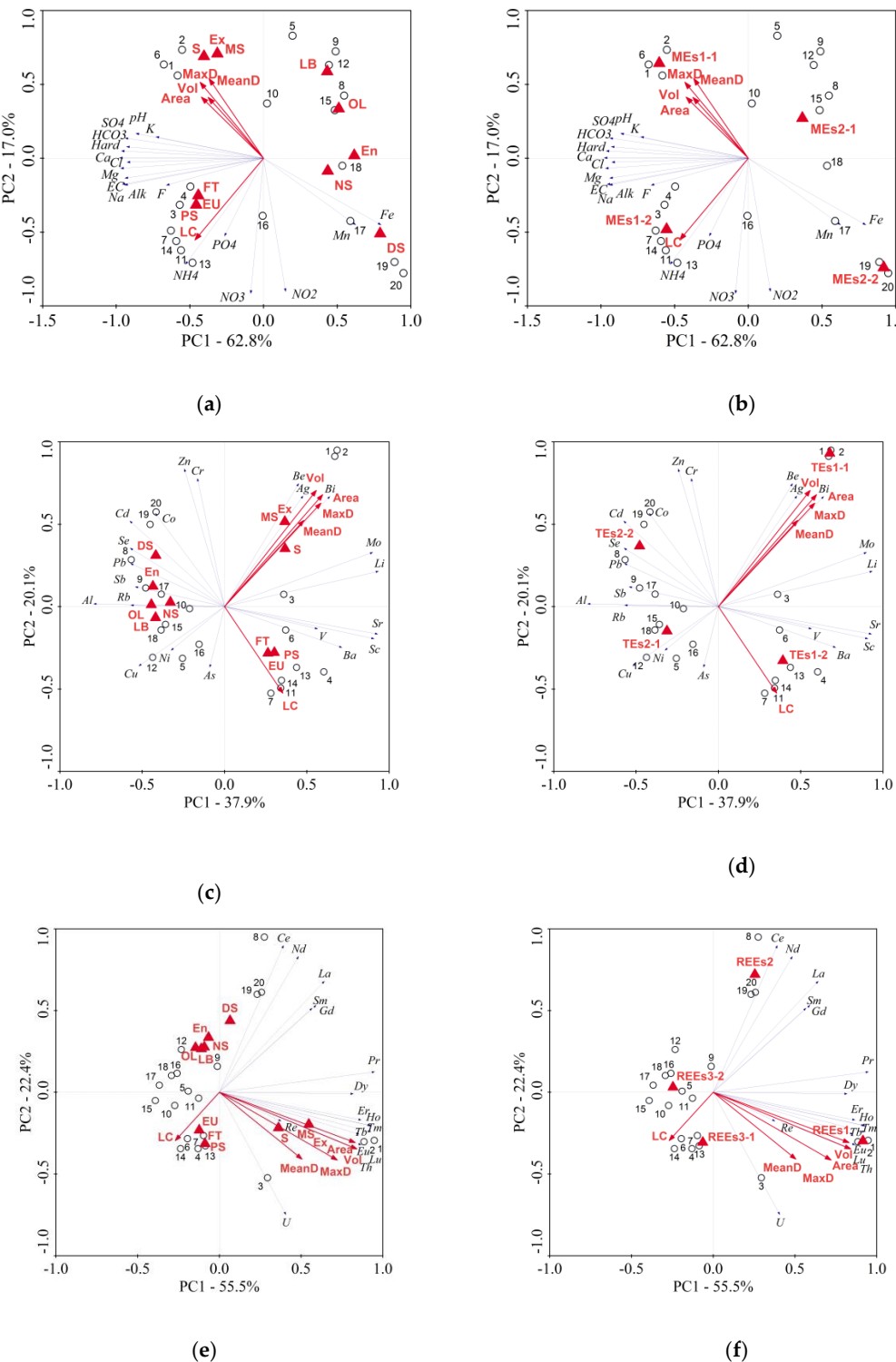

**Figure 6.** Results of principal component analysis (PCA) analysis according to groups identified by CA method and against trophic, thermal and hydrological typology for MEs (**a** and **b**), TEs (**c** and **d**) and REEs (**e** and **f**) (Determination of lakes as in Table 1).

The PCA analysis carried out for MEs, based on the Kaiser criterion, allowed two significant principal components whose eigenvalues were higher than 1 to be distinguished. The components PC1 and PC2 explain respectively 62.8% and 17.0% of the internal data structure. The first main component of PC1 was strongly correlated with the values of pH, EC, Hardness, Alkalinity and $HCO_3^-$ as well as concentrations of $Cl^-$, $SO_4^{2-}$, $Ca^{2+}$, $Mg^{2+}$, $Na^+$. A moderate negative correlation was recorded for $NH_4^+$, F and $K^+$ concentrations. The concentrations of Fe and Mn were positively correlated with PC1 at strong and moderate levels, respectively. There was a negative correlation with the second main component of PC2 at a strong level for $NO_2^-$ and $NO_3^-$ and at a moderate level for $NH_4^+$ and $PO_4^{3-}$ (Figure 6a,b).

Projection of environmental values against the background of macro element concentrations made it possible to determine the relationships between lakes. Eutrophic and dystrophic lakes are characterized by a higher content of nitrogen and phosphorus compounds compared to oligotrophic and mesotrophic lakes. Statistical analysis also showed that lakes with high inflow from the catchment were characterized by the highest values of $NH_4^+$ and $PO_4^{3-}$. Moreover, lobelia lakes are the most similar in terms of ME content to oligotrophic lakes. In this type of lake, the lowest values of all examined MEs were found. The lakes with the lowest depths, surface area and at the same time low capacity—Kacze Oko and Rybie Oko (19 and 20)—were characterized by the highest values of Mn and Fe. The lowest values of Mn and Fe were recorded in Ostrowite (1 and 2) and Zielone (6) lakes (Figure 6a). The obtained results confirmed the earlier classification into groups by the CA method, each of the separated groups being located in a separate part of the coordinate system. The lakes Ostrowite (1 and 2), Zielone (6) (MEs1-1) and Jeleń (3), Bełczak (14), Płęsno (4), Główka (13), Skrzynka (7) and Mielnica (11) (MEs1-2) were characterized by relatively low variability, which is indicated by their small dispersion (Figure 6b). The lakes Krzywce Wielkie (5), Krzywce Małe (10), Olbrachta (16), Gacno Małe (8), Gacno Wielkie (9), Nierybno (12), Głuche (15), Kocioł (17) and Kociołek (18) (MEs2-1) were the most diversified. High similarity of water composition in terms of ME content was found in Kacze Oko (19) and Rybie Oko (20) lakes.

The PCA analysis carried out on the basis of TEs made it possible to distinguish three main components, PC1, PC2 and PC3, whose eigenvalues were greater than 1. The extracted main components explain, respectively, 37.9%, 20.1% and 11.1% of the internal data structure. The concentrations of Sr, Li, Mo and Sc were strongly positively correlated with PC1, while the values of Be, Zn and Cr were strongly correlated with PC2. The dispersion of the analyzed TE shows their high variability in the lakes water. A similarity was observed between the TEs content in the waters of lakes classified as dystrophic and oligotrophic, which are at the same time non-stratified and endorheic lakes (Figure 6c). A strong positive correlation was observed between the surface area, volume and depth of the lakes and the content of Ag, Bi and Be. In eutrophic, partially stratified and flow-through lakes increased values of Ba, Sc, Sr and V were noted. The results of PCA analysis confirmed the results obtained by CA; the TEs1 group was more diverse than the TEs2 group, which is confirmed by the dispersion of points representing particular subgroups (Figure 6d). PCA results for REEs allowed two main components to be selected, PC1 and PC2, which explain, respectively, 55.5% and 22.4% of the internal data structure. Pr, Eu, Tb, Dy, Ho, Er, Tm, Lu and Th were strongly correlated with PC1. Ce and Nd values were strongly correlated with PC2 and negatively with U. Generally, the highest concentrations of REEs were found in Ostrowite Lake, except for La, Ce, Nd, Gd and Sm, of which the highest concentrations were found in Gacno Małe Lake (8), Rybie Oko (20) and Kacze Oko (19). The lowest REE contents were found in oligotrophic and eutrophic lakes (Figure 6e). Moreover, the lowest REE values were found in the lobelia lakes. The analysis showed that with the increase in the lake coefficient, the contents of Ce, Nd, La, Sm and Gd decreased. As in the case of MEs and TEs, the PCA analysis carried out for REEs confirmed the results obtained with the CA method (Figure 6f). The presented issues are related to the global research on the chemical characteristics of lake waters [61–65]. In Poland, it is difficult to identify lakes that are not under the influence of human pressure [66,67]. Pollutants flow directly into the lakes or indirectly by wet and dry atmospheric deposition [6,10,37]. It is therefore essential to

distinguish the influence of anthropogenic factors from natural ones, influencing the chemistry of lake waters. Against this background, the BTNP lakes are located within the national park, the borders of which coincide with the borders of the catchment. The only source of pollution in this case may be the so-called wet and dry deposition. The results are similar to those obtained by Rupakheti et al. [68], who showed that lakes under less human influence are characterized by much better water quality parameters. The content of MEs, TEs and REEs in the BTNP lake waters is mainly due to natural geological processes. Similar results were obtained by Santolaria et al. [6] in the lakes of the Pyrenees, where the main source of the macro and trace elements is geological weathering. Additionally, Wang et al. [59] suggested that atmospheric deposition and weathering of background soils are responsible for a substantial portion of heavy metals in surface water. Wicik and Lenartowicz [69] demonstrated that the BTNP has a mosaic of geochemical landscapes with the dominance of eluvia landscapes. In the BTNP there are subaqueous landscapes, including transaqueous (rivers) and aqueous (lakes). The lakes connected by the Struga Siedmiu Jezior stream are supplied with water mainly from deep levels. The remaining lakes are supplied with shallow groundwater. This is reflected, among others, in the lowest variability of MEs, TEs and REEs found in the lakes connected by the Struga Siedmiu Jezior stream, where the exchange of water with similar physical and chemical characteristics takes place. The similarity results from the long exchange time of water in the lakes [39]. As regards the quality of the lakes in the BTNP, they were examined several times [70,71]. The research shows that in the analyzed period there were no significant changes in the chemistry of lake waters in the area of the BTNP. The issues discussed in the paper refer to these studies, expanding the existing information and establishing a new time cross-section with information on the physical and chemical properties of lake waters in the BTNP. These facts are important for the proper future interpretation of data on the scale and pace of change and, if necessary, for protective action. The authors realize that concentrations of pollutants in water were lower than those in sediment, in addition, the quality of bottom sediments is more stable [39,72]. Therefore, further research will be aimed at analyzing bottom sediments in the BTNP lakes.

## 4. Conclusions

The paper presents the results of chemical analysis of lake waters in the Bory Tucholskie National Park (BTNP). The uniqueness of the BTNP is related to the existence of 24 lakes in a small-scale area and the long-term forest land use. Thus, the lakes have different morphometric parameters, hydrological and trophic types and thermal regimes. Furthermore, the BTNP area is unique due to high variability in geological structure and its location within a single catchment. It eliminates the inflow of pollutants from other anthropogenic areas. The study included 19 lakes, in which 55 parameters of water quality were analyzed, within macro elements (MEs), trace elements (TEs) and rare earth elements (REEs). The analysis of MEs TEs and REEs concentration in relation to the environmental factors and trophic, hydrologic and thermal typology, allowed a better understanding of their spatial distribution in the BTNP lakes. The following detailed conclusions were reached:

- Low concentrations of MEs, TEs and REEs were measured, which confirms the absence of anthropogenic pressure.
- High variation of ME, TE and REE contents were measured between individual lakes due to different geological structure.
- It is possible to divide the studied lakes into six groups taking into account all analyzed parameters of water quality.
- Taking into account the concentrations of MEs and TEs, the lakes were divided into two main groups, each of which was divided into two subgroups. In the case of REEs, three main groups were distinguished; additionally, two subgroups were distinguished in the third group.
- The lobelia lakes were characterized by the lowest concentrations of MEs and REEs, which results mainly from the small catchment area and their mainly endorheic character.

- The highest variability of MEs, TEs and REEs occurred in endorheic lakes, where the geological structure was dominant.
- The lowest variability of MEs, TEs and REEs occurred in the lakes connected by the Struga Siedmiu Jezior stream with the exception of Lake Zielone, which is a specific buffer between Lake Ostrowite and the other lakes.
- The parameters of lakes, i.e., area, depth and volume, were correlated only with ten parameters: Mn, Fe, Bi, Ag, Be, Cu, Ni, Eu, Lu and Th.
- The lake coefficient was correlated with the contents of $NH_4^+$, $PO_4^{3-}$, Co, Gd, Sm and La.

**Author Contributions:** Conceptualization—M.S. (Mariusz Sojka) and A.C.; methodology—M.S. (Mariusz Sojka) and M.S. (Marcin Siepak); data collection—M.P., A.C. and M.S. (Marcin Siepak); chemical analyses—M.S. (Marcin Siepak); statistical analysis and interpretation—M.S. (Mariusz Sojka); writing—original draft preparation, M.S. (Mariusz Sojka); writing—review, M.P. and A.C.; editing manuscript—M.P.; visualization, M.S. (Mariusz Sojka); supervision—M.S. (Mariusz Sojka), A.C and M.P. All authors have read and agreed to the published version of the manuscript.

**Funding:** The publication was co-financed within the framework of Ministry of Science and Higher Education programme as "Regional Initiative Excellence" in years 2019-2022, Project No. 005/RID/2018/19. This research was supported by financial means granted by the Ministry of Science and Higher Education; Research project No. 215862/E-336/SPUB/2017/1.

**Acknowledgments:** The authors would like to thank the Bory Tucholskie National Park Directorate for the possibility of conducting field research and the park employees for their help during its implementation.

**Conflicts of Interest:** The authors declare no conflict of interest.

## Appendix A

**Table A1.** The macro elements (MEs) concentrations in the lake waters of the Bory Tucholskie National Park.

| Parameters | Unit | Sample Point | | | | | | | | | | | | | | | | | | | |
|---|---|---|---|---|---|---|---|---|---|---|---|---|---|---|---|---|---|---|---|---|---|
| | | 1 | 2 | 3 | 4 | 5 | 6 | 7 | 8 | 9 | 10 | 11 | 12 | 13 | 14 | 15 | 16 | 17 | 18 | 19 | 20 |
| WT | °C | 22.4 | 22.9 | 22.2 | 22.3 | 22.5 | 22.0 | 22.5 | 23.6 | 23.4 | 22.2 | 22.2 | 22.8 | 21.8 | 21.3 | 22.6 | 22.1 | 22.2 | 21.6 | 21.6 | 23 |
| pH value | pH | 8.36 | 8.41 | 8.52 | 8.31 | 8.90 | 8.37 | 8.54 | 5.22 | 5.23 | 9.22 | 8.20 | 6.97 | 8.69 | 8.56 | 6.18 | 8.00 | 5.39 | 6.00 | 4.33 | 4.46 |
| EC | µS/cm | 226 | 224 | 231 | 253 | 42.3 | 238 | 241 | 26.9 | 26.4 | 44.2 | 269 | 19.4 | 234 | 226 | 20.1 | 40.6 | 15.6 | 10.7 | 27.6 | 23.2 |
| Alk. | mval/L | 1.4 | 1.4 | 1.5 | 1.7 | 0.2 | 1.5 | 1.8 | 0.2 | 0.2 | 0.3 | 2.1 | 0.2 | 1.6 | 1.4 | 0.2 | 0.6 | 0.2 | 0.3 | 0 | 0 |
| Hard. | mg CaCO$_3$/L | 86.3 | 85.7 | 88.7 | 94.7 | 15.7 | 88.7 | 96.5 | 15.3 | 15.4 | 18.1 | 109.7 | 13.4 | 85.9 | 83.4 | 13.1 | 32.2 | 12.2 | 13.5 | 1.8 | 1.6 |
| HCO$_3^-$ | mg/L | 85.4 | 85.4 | 91.5 | 103.7 | 12.2 | 91.5 | 109.8 | 12.2 | 12.2 | 18.3 | 128.1 | 12.2 | 97.6 | 85.4 | 12.2 | 36.6 | 12.2 | 18.3 | 0 | 0 |
| F$^-$ | mg/L | 0.05 | 0.04 | 0.06 | 0.05 | 0.06 | 0.05 | 0.07 | 0.04 | 0.04 | 0.07 | 0.07 | 0.04 | 0.05 | 0.07 | 0.04 | 0.06 | 0.04 | 0.04 | 0.04 | 0.03 |
| Cl$^-$ | mg/L | 3.95 | 3.84 | 4.02 | 3.68 | 2.51 | 4.05 | 3.69 | 2.50 | 2.41 | 2.80 | 3.57 | 1.85 | 3.74 | 4.06 | 2.09 | 2.54 | 1.69 | 1.53 | 2.11 | 2.02 |
| SO$_4^{2-}$ | mg/L | 17.7 | 17.3 | 16.9 | 12.4 | 4.29 | 17.1 | 11.3 | 4.01 | 4.03 | 3.69 | 10.6 | 2.44 | 13.6 | 15.0 | 2.79 | 2.56 | 1.83 | 0.630 | 0.523 | 0.778 |
| PO$_4^{3-}$ | mg/L | 0.06 | 0.07 | 0.03 | 0.04 | 0.01 | 0.05 | 0.07 | 0.02 | 0.01 | 0.03 | 0.05 | 0.03 | 0.05 | 0.05 | 0.03 | 0.06 | 0.05 | 0.04 | 0.07 | 0.06 |
| NH$_4^+$ | mg/L | 0.121 | 0.111 | 0.319 | 0.353 | 0.226 | 0.160 | 0.399 | 0.088 | 0.073 | 0.152 | 0.382 | 0.070 | 0.442 | 0.429 | 0.082 | 0.379 | 0.131 | 0.176 | 0.228 | 0.196 |
| NO$_2^-$ | mg/L | 0.002 | 0.001 | 0.005 | 0.004 | 0.002 | 0.003 | 0.004 | 0.003 | 0.002 | 0.004 | 0.005 | 0.003 | 0.006 | 0.005 | 0.004 | 0.005 | 0.006 | 0.004 | 0.005 | 0.006 |
| NO$_3^-$ | mg/L | 0.007 | 0.006 | 0.011 | 0.008 | 0.004 | 0.005 | 0.009 | 0.005 | 0.004 | 0.007 | 0.009 | 0.005 | 0.010 | 0.011 | 0.006 | 0.008 | 0.010 | 0.009 | 0.010 | 0.011 |
| K$^+$ | mg/L | 0.41 | 0.39 | 0.32 | 0.30 | 0.07 | 0.80 | 0.47 | 0.10 | 0.18 | 0.36 | 0.37 | 0.26 | 0.23 | 0.28 | 0.24 | 0.32 | 0.16 | 0.24 | 0.14 | 0.11 |
| Na$^+$ | mg/L | 1.99 | 1.96 | 2.17 | 2.26 | 0.66 | 2.26 | 2.25 | 0.44 | 0.73 | 0.86 | 2.37 | 0.59 | 2.35 | 2.12 | 0.57 | 0.74 | 0.52 | 0.57 | 0.57 | 0.57 |
| Mg$^{2+}$ | mg/L | 3.40 | 3.39 | 3.44 | 3.87 | 0.83 | 3.43 | 3.78 | 0.80 | 0.89 | 0.99 | 3.86 | 0.84 | 3.31 | 3.30 | 0.83 | 2.27 | 0.94 | 0.90 | 0.12 | 0.09 |
| Ca$^{2+}$ | mg/L | 29.0 | 28.8 | 29.9 | 31.6 | 4.91 | 29.9 | 32.4 | 4.83 | 4.71 | 5.64 | 37.6 | 3.99 | 29.0 | 28.0 | 3.90 | 9.16 | 3.32 | 3.92 | 0.53 | 0.50 |
| Fe | mg/L | 3.37 | 3.16 | 5.59 | 19.6 | 10.5 | 3.27 | 9.51 | 93.7 | 52.7 | 7.63 | 39.6 | 26.2 | 28.3 | 9.59 | 23.5 | 17.6 | 72.1 | 34.7 | 253.0 | 241.6 |
| Mn | mg/L | 1.64 | 1.73 | 11.7 | 22.3 | 4.93 | 1.87 | 44.8 | 78.6 | 76.9 | 6.37 | 50.6 | 19.6 | 17.0 | 7.45 | 75.2 | 36.0 | 44.0 | 42.5 | 58.9 | 68.3 |

**Table A2.** The trace elements (TEs) concentrations in the lake waters of the Bory Tucholskie National Park.

| Parameters | Unit | Sample Point | | | | | | | | | | | | | | | | | | | |
|---|---|---|---|---|---|---|---|---|---|---|---|---|---|---|---|---|---|---|---|---|---|
| | | 1 | 2 | 3 | 4 | 5 | 6 | 7 | 8 | 9 | 10 | 11 | 12 | 13 | 14 | 15 | 16 | 17 | 18 | 19 | 20 |
| Li | µg/L | 2.78 | 2.70 | 1.58 | 1.85 | 0.61 | 1.77 | 1.79 | 0.41 | 0.48 | 0.75 | 1.90 | 0.36 | 1.28 | 1.18 | 0.48 | 0.52 | 0.49 | 0.45 | 0.74 | 0.84 |
| Be | µg/L | 0.10 | 0.10 | 0.05 | 0.03 | 0.03 | 0.09 | 0.02 | 0.04 | 0.03 | 0.03 | 0.02 | 0.02 | 0.02 | 0.01 | 0.02 | 0.02 | 0.02 | 0.02 | 0.04 | 0.05 |
| Al. | µg/L | 2.25 | 2.31 | 8.58 | 2.58 | 66.9 | 5.80 | 28.2 | 72.8 | 77.3 | 22.5 | 6.90 | 12.7 | 4.75 | 12.5 | 7.65 | 17.1 | 11.8 | 22.7 | 84.8 | 81.2 |
| Sc | µg/L | 0.21 | 0.20 | 0.24 | 0.22 | 0.07 | 0.24 | 0.25 | 0.04 | 0.03 | 0.06 | 0.30 | 0.04 | 0.24 | 0.25 | 0.06 | 0.11 | 0.05 | 0.06 | 0.08 | 0.09 |
| V | µg/L | 0.25 | 0.23 | 0.34 | 0.27 | 0.30 | 0.21 | 0.24 | 0.18 | 0.10 | 0.19 | 0.21 | 0.16 | 0.35 | 0.36 | 0.04 | 0.22 | 0.05 | 0.23 | 0.20 | 0.25 |
| Cr | µg/L | 0.15 | 0.16 | 0.06 | 0.04 | 0.05 | 0.06 | 0.04 | 0.09 | 0.05 | 0.06 | 0.04 | 0.11 | 0.04 | 0.06 | 0.03 | 0.08 | 0.07 | 0.09 | 0.22 | 0.21 |
| Co | µg/L | 0.04 | 0.04 | 0.05 | 0.02 | 0.01 | 0.02 | 0.02 | 0.13 | 0.06 | 0.02 | 0.03 | 0.01 | 0.03 | 0.03 | 0.04 | 0.03 | 0.05 | 0.02 | 0.10 | 0.12 |
| Ni | µg/L | 0.11 | 0.11 | 0.20 | 0.06 | 0.42 | 0.42 | 0.44 | 0.35 | 0.10 | 0.30 | 0.24 | 0.42 | 0.08 | 0.13 | 0.19 | 0.23 | 0.13 | 0.17 | 0.25 | 0.22 |
| Cu | µg/L | 0.49 | 0.47 | 0.91 | 0.43 | 2.52 | 1.43 | 1.72 | 2.45 | 2.42 | 0.90 | 2.51 | 1.82 | 0.67 | 0.75 | 1.21 | 2.18 | 0.91 | 0.96 | 1.08 | 1.01 |
| Zn | µg/L | 15.6 | 15.0 | 10.5 | 1.37 | 3.39 | 4.67 | 2.90 | 12.9 | 14.4 | 11.3 | 3.76 | 4.53 | 5.31 | 3.60 | 7.97 | 3.96 | 6.81 | 6.81 | 10.4 | 11.5 |
| As | µg/L | 0.80 | 0.82 | 0.91 | 0.93 | 1.23 | 0.67 | 0.88 | 1.27 | 0.98 | 0.99 | 0.91 | 2.04 | 1.37 | 1.38 | 0.62 | 0.72 | 0.46 | 1.54 | 0.70 | 0.65 |
| Se | µg/L | 0.27 | 0.26 | 0.24 | 0.09 | 0.28 | 0.20 | 0.24 | 0.20 | 0.18 | 0.36 | 0.14 | 0.34 | 0.21 | 0.16 | 0.34 | 0.34 | 0.29 | 0.38 | 0.41 | 0.39 |
| Rb | µg/L | 0.64 | 0.65 | 0.57 | 0.43 | 0.88 | 0.60 | 0.51 | 0.80 | 0.83 | 1.17 | 0.51 | 0.77 | 0.42 | 0.53 | 1.11 | 1.01 | 1.01 | 0.94 | 0.47 | 0.50 |
| Sr | µg/L | 78.9 | 78.4 | 68.9 | 85.9 | 10.2 | 76.1 | 84.9 | 7.06 | 6.57 | 9.53 | 91.7 | 4.94 | 72.1 | 68.1 | 4.62 | 30.4 | 2.72 | 1.73 | 3.68 | 3.35 |
| Mo | µg/L | 0.61 | 0.63 | 0.36 | 0.32 | 0.14 | 0.26 | 0.24 | 0.12 | 0.10 | 0.11 | 0.21 | 0.10 | 0.33 | 0.26 | 0.07 | 0.08 | 0.06 | 0.06 | 0.05 | 0.07 |
| Ag | µg/L | 0.09 | 0.10 | 0.03 | 0.03 | 0.04 | 0.02 | 0.02 | 0.03 | 0.02 | 0.04 | 0.02 | 0.01 | 0.01 | 0.02 | 0.01 | 0.04 | 0.02 | 0.02 | 0.02 | 0.02 |
| Cd | µg/L | 0.032 | 0.031 | 0.022 | 0.017 | 0.030 | 0.017 | 0.005 | 0.073 | 0.078 | 0.017 | 0.012 | 0.034 | 0.018 | 0.014 | 0.042 | 0.007 | 0.047 | 0.009 | 0.056 | 0.051 |
| Sb | µg/L | 0.18 | 0.17 | 0.16 | 0.08 | 0.25 | 0.09 | 0.07 | 0.29 | 0.23 | 0.17 | 0.07 | 0.44 | 0.10 | 0.15 | 0.17 | 0.18 | 0.11 | 0.27 | 0.12 | 0.16 |
| Ba | µg/L | 5.36 | 5.42 | 5.26 | 6.02 | 2.35 | 5.23 | 5.68 | 4.62 | 4.18 | 2.16 | 8.73 | 4.15 | 6.18 | 5.62 | 2.80 | 2.34 | 0.93 | 1.27 | 1.40 | 0.97 |
| Pb | µg/L | 0.53 | 0.49 | 0.59 | 0.41 | 0.59 | 0.48 | 0.47 | 1.64 | 1.02 | 0.43 | 0.87 | 0.79 | 0.39 | 0.45 | 0.45 | 0.50 | 0.51 | 0.68 | 0.98 | 0.85 |
| Bi | µg/L | 0.32 | 0.33 | 0.11 | 0.07 | 0.06 | 0.07 | 0.05 | 0.04 | 0.03 | 0.05 | 0.04 | 0.03 | 0.05 | 0.04 | 0.02 | 0.05 | 0.03 | 0.03 | 0.03 | 0.04 |

**Table A3.** The rare earth elements (REEs) concentrations in the lake waters of the Bory Tucholskie National Park.

| Parameters | Unit | Sample Point | | | | | | | | | | | | | | | | | | | |
|---|---|---|---|---|---|---|---|---|---|---|---|---|---|---|---|---|---|---|---|---|---|
| | | 1 | 2 | 3 | 4 | 5 | 6 | 7 | 8 | 9 | 10 | 11 | 12 | 13 | 14 | 15 | 16 | 17 | 18 | 19 | 20 |
| La | µg/L | 0.030 | 0.028 | 0.012 | 0.006 | 0.013 | 0.004 | 0.011 | 0.056 | 0.014 | 0.002 | 0.008 | 0.022 | 0.007 | 0.009 | 0.003 | 0.014 | 0.012 | 0.009 | 0.032 | 0.027 |
| Ce | µg/L | 0.031 | 0.030 | 0.014 | 0.013 | 0.019 | 0.007 | 0.010 | 0.082 | 0.034 | 0.015 | 0.017 | 0.040 | 0.008 | 0.010 | 0.019 | 0.032 | 0.018 | 0.023 | 0.071 | 0.072 |
| Pr | µg/L | 0.015 | 0.016 | 0.008 | 0.004 | 0.001 | 0.001 | 0.002 | 0.009 | 0.006 | 0.004 | 0.006 | 0.006 | 0.002 | 0.003 | **0.001** | 0.002 | **0.001** | 0.001 | 0.009 | 0.009 |
| Nd | µg/L | 0.019 | 0.019 | 0.007 | 0.002 | 0.014 | 0.003 | 0.012 | 0.046 | 0.017 | 0.003 | 0.012 | 0.015 | 0.009 | 0.002 | 0.009 | 0.019 | 0.009 | 0.009 | 0.029 | 0.033 |
| Sm | µg/L | 0.010 | 0.009 | 0.002 | 0.003 | 0.002 | 0.002 | 0.004 | 0.006 | 0.008 | **0.001** | 0.008 | 0.006 | 0.005 | 0.002 | **0.001** | 0.002 | 0.001 | 0.006 | 0.016 | 0.018 |
| Eu | µg/L | 0.019 | 0.018 | 0.018 | 0.009 | 0.003 | 0.007 | 0.010 | 0.012 | 0.011 | 0.003 | 0.010 | 0.003 | 0.007 | 0.003 | 0.002 | 0.001 | 0.001 | 0.001 | 0.007 | 0.006 |
| Gd | µg/L | 0.011 | 0.010 | 0.004 | 0.004 | 0.006 | 0.006 | 0.001 | 0.019 | 0.007 | 0.009 | 0.007 | 0.003 | 0.004 | 0.003 | 0.002 | 0.001 | 0.006 | 0.005 | 0.009 | 0.009 |
| Tb | µg/L | 0.013 | 0.012 | 0.008 | 0.001 | **0.001** | 0.001 | 0.004 | 0.005 | 0.003 | 0.001 | **0.001** | **0.001** | 0.001 | **0.001** | **0.001** | **0.001** | **0.001** | 0.001 | 0.002 | 0.002 |
| Dy | µg/L | 0.008 | 0.007 | 0.003 | 0.002 | 0.001 | 0.001 | 0.001 | **0.001** | **0.001** | **0.001** | 0.001 | **0.001** | 0.003 | 0.001 | **0.001** | 0.003 | **0.001** | 0.002 | 0.006 | 0.006 |
| Ho | µg/L | 0.014 | 0.013 | 0.007 | 0.002 | 0.001 | 0.002 | 0.003 | 0.006 | 0.004 | 0.001 | 0.002 | 0.001 | 0.004 | 0.001 | 0.002 | 0.001 | 0.001 | 0.001 | 0.002 | 0.002 |
| Er | µg/L | 0.013 | 0.011 | 0.007 | 0.004 | 0.008 | 0.004 | 0.006 | 0.004 | 0.004 | 0.003 | 0.003 | 0.003 | 0.004 | 0.002 | 0.003 | 0.001 | 0.001 | 0.004 | 0.006 | 0.006 |
| Tm | µg/L | 0.018 | 0.017 | 0.006 | 0.002 | 0.002 | 0.002 | 0.004 | 0.005 | 0.004 | 0.001 | 0.002 | 0.001 | 0.003 | 0.001 | 0.003 | 0.002 | 0.001 | 0.002 | 0.004 | 0.004 |
| Lu | µg/L | 0.012 | 0.011 | 0.006 | 0.003 | 0.002 | 0.002 | 0.002 | 0.003 | 0.006 | 0.004 | 0.002 | 0.002 | 0.004 | 0.002 | 0.002 | 0.002 | 0.003 | 0.002 | 0.002 | 0.003 |
| Re | µg/L | 0.214 | 0.278 | 0.356 | 0.356 | 0.249 | 0.142 | 0.285 | 0.142 | 0.107 | 0.129 | 0.249 | 0.214 | 0.249 | 0.392 | 0.036 | 0.142 | 0.059 | 0.059 | 0.357 | 0.356 |
| Th | µg/L | 0.035 | 0.036 | 0.026 | 0.015 | 0.013 | 0.021 | 0.011 | 0.007 | 0.009 | 0.010 | 0.012 | 0.006 | 0.016 | 0.012 | 0.003 | 0.016 | 0.009 | 0.004 | 0.019 | 0.021 |
| U | µg/L | 0.139 | 0.142 | 0.140 | 0.130 | 0.007 | 0.077 | 0.098 | 0.005 | 0.004 | 0.006 | 0.081 | 0.007 | 0.159 | 0.156 | 0.008 | 0.007 | 0.004 | 0.003 | 0.006 | 0.006 |

Bold value below limit of detection <0.001.

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
