# Peer review of "The Variability of Lake Water Chemistry in the Bory Tucholskie National Park (Northern Poland)"

_water, doi:10.3390/w12020394_

Round 1

Reviewer 1 Report

The paper investigates water chemistry variability in the lakes of the Bory Tucholskie National Park, showing how the highest variability occurred in endorheic lakes, while the lowest occurred in the lakes connected by the Struga Siedmiu Jezior stream. I found the case study and the results interesting and the paper is overall well written. The manuscript can be accepted for publication in Water after an appropriate revision.

The innovation of the study should be highlighted in terms of new findings for environmental studies especially as regards abstract and conclusions. The written English should be carefully revised in terms of repetitions, grammar and sentence structure. Please check the use of abbreviations and acronyms, once defined the first time in the text please use them consistently in the rest of the manuscript. I suggest to add to figure 1 (right) the location of sampling sites and the rivers mentioned in the text to help the reader. I think a brief summary of the paper is needed in the conclusions just before the bullet list.

Author Response

The paper investigates water chemistry variability in the lakes of the Bory Tucholskie National Park, showing how the highest variability occurred in endorheic lakes, while the lowest occurred in the lakes connected by the Struga Siedmiu Jezior stream. I found the case study and the results interesting and the paper is overall well written. The manuscript can be accepted for publication in Water after an appropriate revision.

The authors thank very much for the insightful reviews that have increased the scientific value of the paper. The authors responded to all comments and suggestions.

The innovation of the study should be highlighted in terms of new findings for environmental studies especially as regards abstract and conclusions.

The Abstract and Conclusion sections have been completed.

The written English should be carefully revised in terms of repetitions, grammar and sentence structure.

The English language has been carefully revised in terms of repetitions, grammar and sentence structure.

Please check the use of abbreviations and acronyms, once defined the first time in the text please use them consistently in the rest of the manuscript.

Abbreviations and acronyms have been carefully checked throughout the whole paper.

I suggest to add to figure 1 (right) the location of sampling sites and the rivers mentioned in the text to help the reader.

The location of measurement points in Figure 1 has been supplemented.

I think a brief summary of the paper is needed in the conclusions just before the bullet list.

The Conclusion section was complemented by a brief summary and then detailed conclusions are presented.

Reviewer 2 Report

In my opinion, the publication was prepared in a very good way. Authors described in detail and in detail the issue they addressed. The obtained results were discussed against the background of world literature. The presented results can be very interesting for other people dealing with water quality issues.

Author Response

In my opinion, the publication was prepared in a very good way. Authors described in detail and in detail the issue they addressed. The obtained results were discussed against the background of world literature. The presented results can be very interesting for other people dealing with water quality issues.

The authors thank very much for the insightful reviews that have increased the scientific value of the paper.

Round 2

Reviewer 1 Report

The authors have addressed all my previous concerns in a satisfactory way. I recommend acceptance.